# 3D Composite PDMS/MWCNTs Aerogel as High-Performing Anodes in Microbial Fuel Cells

**DOI:** 10.3390/nano12234335

**Published:** 2022-12-06

**Authors:** Giulia Massaglia, Marzia Quaglio

**Affiliations:** 1Department of Applied Science and Technology, Politecnico di Torino, 10129 Torino, Italy; 2Center for Sustainable Future Technologies@ POLITO, Istituto Italiano di Tecnologia, 10100 Torino, Italy

**Keywords:** microbial fuel cells, nanostructured anode, 3D composite aerogel, polydimethylsiloxane, multi-wall carbon nanotubes

## Abstract

Porous 3D composite materials are interesting anode electrodes for single chamber microbial fuel cells (SCMFCs) since they exploit a surface layer that is able to achieve the correct biocompatibility for the proliferation of electroactive bacteria and have an inner charge transfer element that favors electron transfer and improves the electrochemical activity of microorganisms. The crucial step is to fine-tune the continuous porosity inside the anode electrode, thus enhancing the bacterial growth, adhesion, and proliferation, and the substrate’s transport and waste products removal, avoiding pore clogging. To this purpose, a novel approach to synthetize a 3D composite aerogel is proposed in the present work. A 3D composite aerogel, based on polydimethylsiloxane (PDMS) and multi-wall carbon nanotubes (MWCNTs) as a conductive filler, was obtained by pouring this mixture over the commercial sugar, used as removable template to induce and tune the hierarchical continuous porosity into final nanostructures. In this scenario, the granularity of the sugar directly affects the porosities distribution inside the 3D composite aerogel, as confirmed by the morphological characterizations implemented. We demonstrated the capability to realize a high-performance bioelectrode, which showed a 3D porous structure characterized by a high surface area typical of aerogel materials, the required biocompatibility for bacterial proliferations, and an improved electron pathway inside it. Indeed, SCMFCs with 3D composite aerogel achieved current densities of (691.7 ± 9.5) mA m^−2^, three orders of magnitude higher than commercial carbon paper, (287.8 ± 16.1) mA m^−2^.

## 1. Introduction

Alternative energy sources and technologies have progressively gained a crucial role in the replacement of traditional fossil energy, enhancing the consequent environmental healthiness [1,2]. In this scenario, bio-electrochemistry devices (BESs) can satisfy all requirements of renewable energy and wastewater treatment, since these devices are able to directly transduce the chemical energy contained in an organic mass into an electrical energy, thanks to the metabolic activity of electroactive bacteria [3,4]. Electroactive bacteria have been exploited for different technologies, such as renewable energy generation [5], wastewater treatment [6], biosensors [7], and bioremediation [8]. The main bio-electrochemistry device that involves electroactive bacteria to obtain a power output is known as a microbial fuel cell (MFC). The peculiarity of these devices is the potential for a wide range of applications, such as in biosensors, bioremediation, wastewater treatment, and energy generation.

In the last decades, many works in the literature focused their attention on all the possible efforts to improve the MFCs’ performance in terms of power output, which results to be very low, leading thus to a discrepancy between the prospective technology and real-world applications [9]. Among all possible technological components, such as electroactive bacteria species, device architecture, ion exchange membrane, and organic substrates, bioelectrodes play a pivotal role in defining the MFCs’ performance [10,11]. Bioelectrodes, indeed, result to be the solid substrate on which the electroactive bacteria can proliferate, thus inducing biofilm formation and the consequent exchange of electrons [12]. To guarantee these features, the physiochemical properties and structures of electrodes are key factors that strictly affect the electron transfer at the biological/inorganic interface and define the maximum available surface area for the electroactive bacteria’s attachment and growth [10,11,12]. Carbon-based materials are considered one of the best-performing anode electrodes, able to combine the biocompatibility properties for the proliferation of microorganisms and the proper electrical conductivity, thus ensuring electron transfer from electroactive bacteria and anode electrodes [9,10,12,13,14]. The lower electrical conductivity of carbon-based materials, due to the single metal electrode, is widely balanced by their capability to improve bacterial proliferation, inducing an optimized biofilm formation [15,16]. Moreover, carbon-based materials exploit many other important properties, such as lightweight, cost-effective electrodes, a high surface area to volume ratio, a chemically inert surface, and proper mechanical features, giving them all the required features of a well-performing anode electrode [17]. Previous studies applied different carbon materials as anode electrodes, such as carbon cloth, carbon paper graphite rod, carbon sponges, and metal oxide foam with conductive coatings [18,19]. Among them, a particular interest for porous electrodes has arisen due to the high surface area capable of improving the growth, adhesion and proliferation of bacteria. Nevertheless, pore clogging frequently occurs during the phase of biofilm formation, when the porous electrodes are implemented [18,19]. Consequently, the transport of substrates and waste products inside 3D electrodes are largely limited, reflecting a low MFC performance. The random, long-distance, zig-zag ion migration paths of disordered electrode porous structure can hinder the rate of substrate transport and waste removal [20,21].

Several works in the literature investigated the fabrication of customizable 3D electrodes, characterized by a defined geometric structure and known channel sizes, confirming the potential for fabricating electrodes via 3D printing technology. This technology, however, is not sufficient to completely overcome the main limits of 3D electrodes, such as the low specific area available for bacteria adhesion, maintaining a low power output density [22,23,24,25]. Furthermore, in recent years, composite anode electrodes have attracted great interest [17], since these electrodes are characterized by two interconnected elements/portions: a polymeric layer suitable for creating the correct biocompatibility for bacterial proliferation and a conductive filler capable of enhancing the electron transfer from electroactive bacteria to the anodic surface. In this scenario, several works in the literature [26,27,28,29,30,31,32,33,34] focused their attention on the development of composite anodes, which involved an intrinsic conductive polymer, such as polyaniline (PANI) [26,27], polypyrrole [28], or carbon black/stainless steel mesh composite electrode [29] as a surface layer, while carbon nanotubes (CNTs) played a key role not only in improving electron transfer from the microorganisms to the electrode surface, but also in inducing an improved nanostructured habitat able to increase bacterial growth, as discussed in the literature [30,31,32,33,34,35]. During the last years, furthermore, CNTs showed an emerging and interesting application as anodes in MFCs since they exhibit unique properties in terms of structural features, and mechanical, electrical, physical, and chemical characteristics [31,32]. As demonstrated in the literature [32,33,34], CNTs were shown to be fundamental to the improvement of overall MFC performance thanks to their capability to enhance the electrochemical activity of microorganisms [32] and to create a proper electron pathway, thus enhancing their transfer from microorganisms to the anode surface.

Concerning the possible implementation of a 3D porous composite electrode with a high surface area, which is suitable to improve the overall device performance, the main aim of the present work was to implement a 3D porous composite anode, based on polydimethylsiloxane (PDMS) and multi-wall carbon nanotubes (MWCNTs). In this scenario, PDMS was involved as a biocompatible and flexible layer, able to improve bacterial proliferation on the anode electrode, showing however, an insulating behavior from an electrical point of view, while MWCNTs were employed as a conductive filler able to create proper electron pathways to optimize the electron transfer from electroactive bacteria to the anode surface, granting also the anode electrodes their unique properties, and thus improving the electrochemical activity of the microorganisms [32]. In the present work, commercial sugar was used as a removable template to induce and modulate a continuous porosity inside the material, giving it all the features of aerogel materials, such as a very light weight, low bulk density, high porosity, and a consequent very high specific surface area, leading to a final 3D-like composite aerogel [36,37].

We demonstrate a high-performance bioelectrode, which showed a 3D porous structure characterized by a high surface area typical of aerogel materials, combined with the composite features. Concerning the MFCs’ overall performance reached with composite 3D-like aerogel, a maximum current density of (691.7 ± 9.5) mA m^−2^, which is three orders of magnitude higher than commercial carbon paper anodes (close to (287.8 ± 16.1) mA m^−2^), was obtained. With the main purpose of demonstrating that overall performance reached with *3D composite aerogel* could be comparable with several optimized composite anode electrodes based on CNTs, a volumetric power density was defined by normalizing the maximum achieved power output for the internal volume of our device (equal to 12.5 mL). SCMFCs with *3D composite aerogel* reached a maximum volumetric power density of (3.98 ± 0.06) W m^−3^, resulting in a comparable performance to that achieved with various CNT-based anode materials, as summarized in Table 1.

## 2. Methods and Materials

### 2.1. Materials and Nanofibers Synthesis

With the main aim of synthetizing a 3D composite-like aerogel, a commercial sugar (refined form of sucrose, primary example of disaccharide) was involved as the template to generate a hierarchical porous structure inside the materials. With this strategy, we proposed two different samples: *(i) 3D PDMS aerogel*, obtained by pouring over the sugar a mixture of PDMS (purchased from Sylgard 184, Dow Corning Co., Midland, MI, USA) and its crosslinker with a weight ratio of 10:1, representing the standard ratio for both components, to provide desirable mechanical properties in terms of flexibility and an optimum biocompatibility for electroactive bacteria growth; and *(ii) 3D composite aerogel* obtained by pouring over the sugar a mixture containing PDMS, its crosslinker, and a certain amount of multi-wall carbon nanotubes (MWCNTs) (obtained from Nanocyl).

In this case, the minimum amount of MWCNTs, close to 10 wt%, defined with respect to the amount of commercial elastomer PDMS, was added to PDMS mixed with its crosslinker (weight ratio of 10:1). The amounts of MWCNTs were selected to obtain a proper electrical conductivity of final materials, in line with all works reported in the literature [26,27,28,29,30,31,32,33,34,35].

The mixture was properly stirred for a couple of minutes and the resulting air-bubbles, trapped during the mixture, were removed by imposing gentle vacuum conditions. This composite-based mixture was subsequently poured over the sugar, and it was infiltrated into the porous structure by capillary force. Consequently, the size of the pores and porosity were directly affected and tuned by the granularity of sugar. The final *3D composite aerogel* was obtained through a curing process, implemented in ambient condition at a temperature of 100 °C for 15 min, and the sugar was subsequently dissolved in hot water bath under sonication for 1 h. Finally, the *3D composite aerogel* was dried overnight at room temperature.

### 2.2. Characterizations and Measurements

Morphological properties were evaluated by implementing a field effect scanning electron microscope (FESEM, ZEISS Merlin, Milan, Italy). With the main goal of defining how the granularity of sugar and the presence of conductive filler can affect the hierarchical distribution of porosity inside the materials, FESEM images were processed with imaging software (ImageJ, online applet, National Institutes of Health NIH,) Bethesda, (US-MD). As reported in our previous work [38], final porosity of all samples was defined taking into account the density of 3D composite aerogel $\rho_{3D-areogel\;}$and the density of a bulk substrate made of the same materials, $\rho_{bulk}$, according to Equation (1)
(1)1−ρ3D−areogelρbulk∗100

We compared the porosity distribution for all obtained samples, *3D PDMS aerogel* compared with *3D composite aerogel*, with the main purpose of verifying the effect of MWCNTs, introduced as a conductive filler, on porosity distribution, leading also to the evaluation of how the porosity can affect the lightening properties of samples.

The electrical conductivity of *3D PDMS aerogel* and *3D composite aerogel* was measured to confirm the pivotal role of MWCNTs in increasing the electrical conductivity, which must be such as to allow its application as an anodic electrode. All electrical characterizations (Keithley 2635A, multimeter unit) were performed by employing different voltage values (V) among the samples, leading to measurement of the related current (I) through the material. To implement electrical characterizations, a specimen with a form of a rectangular parallelepiped was defined for two samples, *3D composite aerogel* compared with *3D composite bulk*, which presents the same polymeric composition without porosity distribution inside it. Both of the two materials were compared with a commercial carbon-based material (carbon paper, CP), commonly showing high performance in terms of electrical conductivity.

### 2.3. SCMFC Devices and Working Operation

An open-air single chamber MFC (SCMFC) was used as thoroughly reported in our previous work [39,40]. The MFCs’ devices were fabricated by 3D printing technology (OBJET 30), and they were based on an intermediate compartment that maintained a constant distance between anodic and cathodic parts. The single chamber configuration provided for a common electrolyte, containing 1 g L^−1^ of sodium acetate used as a carbon energy source and other compounds suitable for the optimal operation of these devices. All these compounds were based on ammonium chloride (0.31 g L^−1^ of NH_4_Cl) used as a nitrogen source to ensure bacterial growth, and phosphate buffer solution (PBS) able to maintain a neutral pH (based on 0.13 g L^−1^ of potassium chloride, 4.28 g L^−1^ of sodium phosphate dibasic, and 2.45 g L^−1^ of sodium phosphate monobasic monohydrate). The inner volume of electrolyte was equal to 12.5 mL, while both electrodes showed a geometric area close to 5.76 cm^2^.

With the main aim of evaluating the SCMFCs’ performance, two different anodes electrodes were proposed: *3D composite aerogel* based on PDMS/MWCNTs compared with a commercial carbon-based material (carbon paper (CP) purchased from Fuel Cell Earth), used as reference material. For all devices, commercial cathode electrodes were employed, as proposed in the literature [40,41]. Specifically, the cathode was made of carbon-based material, which showed on the outer side a gas diffusion layer, able to ensure oxygen diffusion from outside to inside the device, and on the inner side, a platinum-based catalyst was properly deposited. This catalyst layer was a conductive paste of platinum (Pt/C, 0.5 mg/cm^2^, from Sigma Aldrich, Burlington, MA, USA) and 5 wt% of Nafion (Sigma Aldrich, Burlington, MA, USA) was used as a binder. Titanium wires (Goodfellow Cambridge Limited, Huntingdon PE29 6WR, United Kingdom UK) were threaded along both anode and cathode electrodes to establish good electrical contact.

To obtain sufficient data for statistical analysis, we implemented 2 SCMFCs for each anode electrode; 2 SCMFCs with 3D composite aerogels and 2 SCMFCs with CP, used as a reference anode, were studied. Another important aspect of these experiments was the electroactive bacteria involved, which were a mixed consortium derived from seawater sediment. All devices worked under fed-batch mode, in which all electrolyte was replaced with new when a voltage value tending to zero, was registered/recorded. Moreover, to evaluate the devices’ overall performance, anodes and cathodes were connected to an external load of 1000 to Ω, and the voltage trends were continuously measured using a multi-channel data acquisition unit (Agilent 34972A, Leini, Italy). With the main purpose of evaluating how 3D composite aerogel can affect the internal resistances of the SCMFCs, electrochemical impedance spectroscopy (EIS) was implemented on all whole devices. EIS was performed by superimposing a sinusoidal signal with an amplitude of 25 mV in the frequency range from 200 mHz to 150 kHz. This electrochemical characterization, moreover, was implemented when all whole devices produced the maximum voltage output, corresponding to an open-circuit voltage (OCV) condition [40,41,42].

## 3. Results and Discussion

In the present work, *3D composite aerogels* based on PDMS and MWCNTs were fabricated by involving a commercial sugar as the template to generate a hierarchical porous structure inside the materials, giving them the intrinsic properties of aerogels, such as high porosity, low density, and light weight. A standard ratio of 10:1 of PDMS and its curing agent was implemented, providing the desirable mechanical properties and an optimum biocompatibility for electroactive bacteria. Moreover, a minimum amount of 10 wt% of MWCNTs was added to reach the proper electrical conductivity [26,27,28,29,30,31,32,33,34,35] to guarantee an effective electron transfer from the bacteria to the anode surface. The 3D composite aerogels allowed for the combination of all properties required to reach an anode material suitable for improving the electrochemical activity of the microorganisms, thus enabling a biofilm formation where the electroactive bacteria can directly transfer the produced electrons to the anode surface (see Figure 1).

Indeed, PDMS was selected for its high biocompatibility to ensure bacterial proliferations, and good chemical and mechanical stabilities, making it suitable to apply as anodic electrodes in SCMFCs. Moreover, the selection of MWCNTs as the conductive filler was made to ensure a proper electrical conductibility of the 3D composite aerogel, which must guarantee a proper electron transfer from the biofilm to the anode electrode surface inside the bio-electrochemical devices.

### 3.1. Morphological Characterization

With the main aim of demonstrating the capability to obtain a 3D porous electrode combined with a high surface area, a morphological characterization was performed. Figure 2 reports electron microscopy characterizations and morphological properties allowing for the evaluation and comparison of the porosity distribution for each of the 3D aerogels, *3D PDMS aerogels* (Figure 2a) compared with *3D composite aerogels* (Figure 2b). Significant differences in pore distributions between two samples can be highlighted by analyzing the same figures. Indeed, *3D composite aerogels* (Figure 2b) are characterized by a hierarchical continuous porosity, suitable for ensuring a high surface area to volume ratio, playing a pivotal role in improving bacterial growth, and simultaneously avoiding pore clogging, a phenomenon that can occur when 3D porous electrodes are used in bio-electrochemical devices. Moreover, *3D composite aerogels* showed pores with sizes in the range of several tens of micrometers (56±13 µm), which considerably affects and favors microorganism proliferation and penetration inside the samples differently from the *3D PDMS aerogels*, which showed a lower porosity distribution inside. All morphological properties, highlighted by FESEM pictures, were confirmed by the definition of porosity percentage, defined by analyzing all images and applying Equation (1), as reported in Figure 2c.

It was possible to observe the correlation between the porosity distribution, which increased when MWCNTs were added, and the density of final samples, whose trend was shown to be opposite, decreasing with the addition of MWCNTs because of the increased porosity. The *3D composite aerogels* were characterized by a porosity distribution that was double that obtained with *3D PDMS aerogels*, ensuring a decreasing of the density and confirming a final 3D sample lighter than the 3D PDMS aerogel.

### 3.2. Electrical Characterization

A careful evaluation of the final electrical conductivity of *3D composite aerogel* was carried out to determine how the aerogel structures can affect the electrical conductivity compared with the one obtained with the *3D composite bulk* (PDMS/MWCNTs) sample. In the present work, to perform the electrical characterization, a specimen in the form of a rectangular parallelepiped was defined, and all characterizations were implemented onto 10 samples for each structure, aerogel, and bulk, respectively. The specimen was characterized by a length of 2 cm, a depth of 0.4 cm, and a thickness of 0.1 cm. Furthermore, with the main purpose of comparing the electrical conductivity obtained with a 3D composite bulk and a *3D composite aerogel* containing PDMS as polymeric matrix and 5 wt% of MWCNTs as conductive filler, a specimen with the same dimensions was used. Figure 3 shows how the aerogel structure presented a higher electrical conductivity, close to (26.5 ± 2.8) mS cm^−1^, which results to be two times higher than that of (10.6 ± 2.5) mS cm^−1^ obtained with *3D composite bulk* samples. However, on the contrary, 3D composite electrical conductivity was found to be comparable with that reached when carbon paper (CP, (23.6 ± 2.5) mS cm^−1^) was used as the reference anode electrode; this is commonly characterized by high performance in terms of electrical conductivity.

These latter results, further confirmed the quality of results in terms of electrical conductivity, leading thus to open the doors for *3D composite aerogel* as anode electrodes in SCMFCs

### 3.3. Performance of Bio-Electrochemical Devices

All of the above results confirmed that *3D composite aerogels* are a good candidate to be applied as anodic electrodes for SCMFCs, since they satisfy the mandatory properties that an anode electrode should have for this application. Furthermore, all results demonstrated that *3D composite aerogel* showed a high continuous porosity, which is suitable to improve the bacteria proliferation and mass transport, and good electrical conductivity, which is important to guarantee an electrical pathway for the electrons released from electroactive microorganisms. The overall performance reached with *3D composite aerogels* as anode electrodes was compared with carbon-based materials (carbon paper), used as reference material. Contrarily, due to the limits attributed to the low porosity of anode electrodes thoroughly investigated by many works in the literature [16,38], in the present work, 3D composite bulk material was not applied as an anode electrode.

Figure 4 demonstrates the analysis of all current density trends obtained with all SCMFCs devices. The current density was defined by normalizing the current values with respect to the geometric area, equal to 5.76 cm^2^.

The maximum current density reached with *3D composite aerogel*, (691.7 ± 9.5) mA m^−2^, was almost three orders of magnitude higher than that of the carbon paper anodes used as the reference anode electrode (close to (287.8 ± 16.1) mA m^−2^). Moreover, with the main purpose of demonstrating the effective role of *3D composite aerogel* in improving the overall SCMFCs performance, a volumetric power density was defined by normalizing the maximum achieved power output for the internal volume, equal to 12.5 mL. SCMFCs with 3D composite aerogel reached a maximum volumetric power density of (3.98 ± 0.06) W m^−3^, which was comparable with the performance achieved with various CNT-based anode materials [26,27,28,29,30,31,32,33,34,35], as reported in Table 1.

Improved overall SCMFC performance, defined as current density trends monitored over time when *3D composite aerogel* was involved, was confirmed by polarization curves obtained by implementing linear sweep voltammetry (LSV). Figure 5 displays the maximum power density, close to 60 mW m^−2^, reached by *3D composite aerogel*, which was three times higher than that obtained with the reference anode electrode, close to 20 mW m^−2^.

The same considerations can be made by analyzing short-circuit current densities, equal to 200 mA m^−2^ for *3D composite aerogel* and equal to 50 mA m^−2^ for reference anode electrodes.

Furthermore, with the main purpose of demonstrating how the continuous porosity combined with good electrical conductivity played a pivotal role in creating an optimal habitat for electroactive bacterial growth, favoring microorganism penetration into the electrode, and improving the feed’s diffusion rate, electrochemical impedance spectroscopy was performed to thoroughly investigate all electrochemical interfaces. EIS was performed to define the internal resistance (R_ct_) related to the charge-transfer of different anode electrodes. The Nyquist plot, represented in Figure 6, demonstrated that SCMFCs having 3D composite aerogel as anode electrodes were characterized by a lower impedance value than those obtained with carbon paper used ed as anodic reference material. As reported in the literature [42], through the analysis of Nyquist plots of impedance obtained for SCMFCs, four different characteristics can be defined: *(i)* ohmic resistance, *(ii)* charge-transfer resistance defining the cathode interfaces created by the presence of the catalyst layer, *(iii)* charge-transfer resistance due to the anode interfaces, and *(iv)* resistance related to diffusion in the electrolyte solution. Since in this experimental configuration, all SCMFCs contained the same materials except for the anode electrode, it was possible to declare that a lower total cell impedance can be attributed to the presence of *3D composite aerogel* as the anode electrode in SCMFCs. The value of R_ct_ for *3D composite aerogel*, equal to 30 Ω, was quite smaller than that obtained with CP as the anode electrodes, which was equal to 47 Ω. A smaller R_ct_ value corresponds to good electron transfer, ensured by the presence of a conductive filler, leading thus to exploit the active role of MWCNTs.

## 4. Conclusions

In the present work, the crucial role of improved anode electrodes to enhance the performance of MFC was demonstrated. It was possible to confirm the key role of a 3D-composite anode electrode, exploited through the development of *3D composite aerogels*. We demonstrated the capability to induce continuous porosity inside the final material by pouring a composite mixture, based on PDMS, its curing agent, and 10 wt% of MWCNTs onto the sugar, which acted as a removal template. To this purpose, *3D composite aerogels* combined a high continuous porosity (close to 60% of whole sample), a light weight ensured by a low density, and a good electrical conductivity of (26.5 ± 2.8) mS cm^−1^.

3D composite *aerogels* have huge potential as anode electrodes in SCMFCs, since these kinds of materials satisfy all those properties, which are deemed mandatory for anode electrodes, such high continuous porosity, a high surface area capable of improving bacterial growth, adhesion, and proliferation, and a good electrical conductivity to offer a proper pathway for electrons from the microorganisms to the outside. SCMFCs with *3D composite aerogels* achieved a maximum current density of (691.7 ± 9.5) mA m^−2^, three times higher than that reached with a commercial carbon-based electrode. Furthermore, this value was reached by a high-performing anode, suitable for inducing biofilm formation derived from an environmental inoculum, leading thus to ensure a decreasing cost of synthesis and an easier fabrication process.

## Figures and Tables

**Figure 1 nanomaterials-12-04335-f001:**
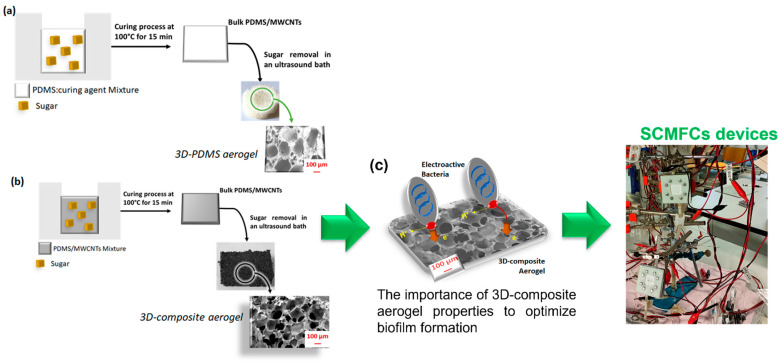
Representation of the scheme implemented to synthetize: (**a**) *3D PDMS aerogel*, obtained by pouring the sugar over the mixture of PDMS and its crosslinker with a weight ratio of 10:1; (**b**) *3D composite aerogel*, based on PDMS and 10 wt% MWCNTs, involving sugar as a porosity template and starting from a bulk layer of PDMS and MWCNTs; (**c**) scheme of the importance of 3D composite aerogel properties in the optimization of biofilm formation, as reported in the literature [43] and a real figure of SCMFC devices, where 3D composite aerogel was applied as anode electrode.

**Figure 2 nanomaterials-12-04335-f002:**
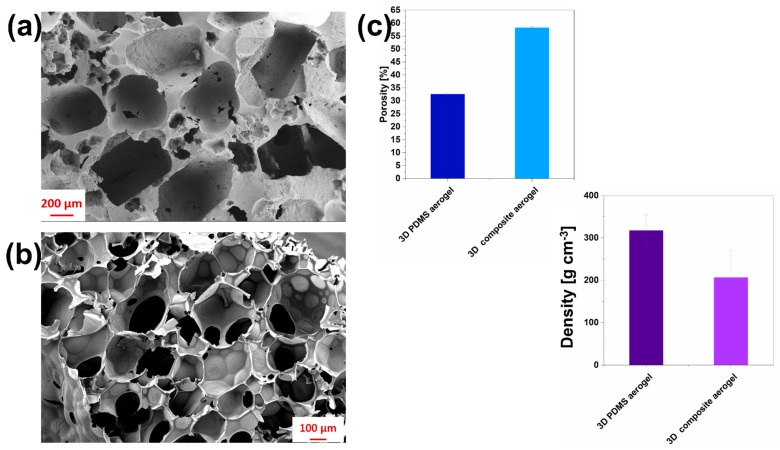
Morphological properties of (**a**) *3D PDMS aerogels*; (**b**) *3D composite aerogel*, based on PDMS/MWCNTs; and (**c**) density and porosity trends characterizing all samples, *3D PDMS aerogels* and *3D composite aerogels*, respectively.

**Figure 3 nanomaterials-12-04335-f003:**
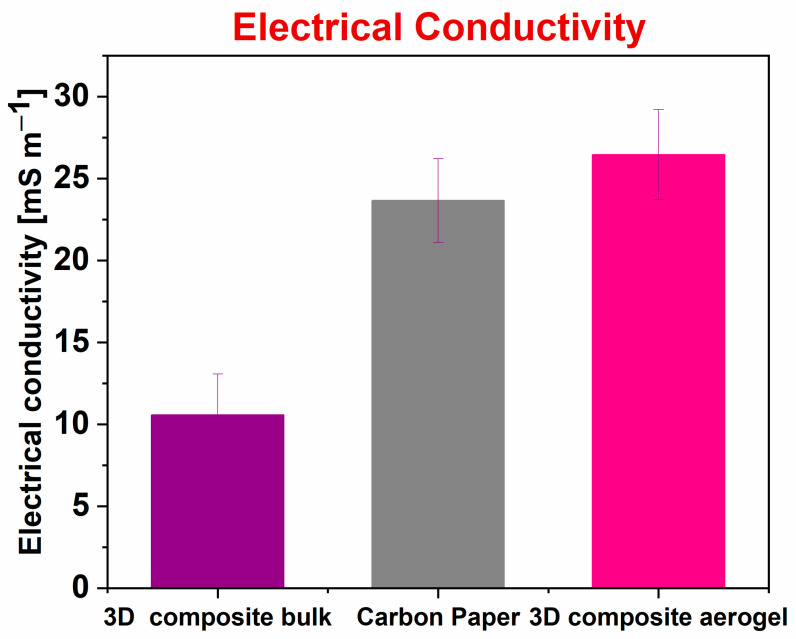
Electrical conductivity obtained for each sample: 3D composite bulk (purple rectangle), carbon paper (CP) (grey rectangle), and *3D composite aerogel* (magenta rectangle). The 3D composite bulk and *3D composite aerogel* were both made of PDMS and 5 wt% of MWCNTs. To obtain these results, all electrical characterizations were performed on 10 samples for each structure.

**Figure 4 nanomaterials-12-04335-f004:**
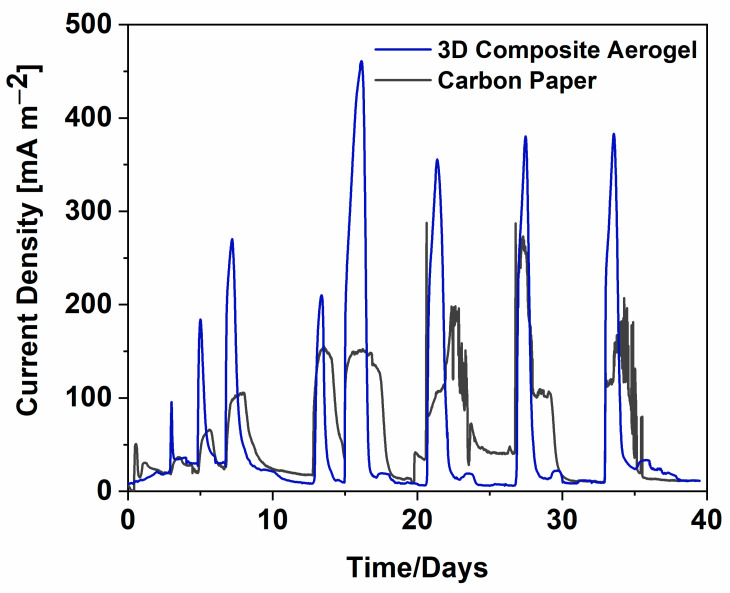
Current density trends represented as a function of time and normalized with respect to the geometric area.

**Figure 5 nanomaterials-12-04335-f005:**
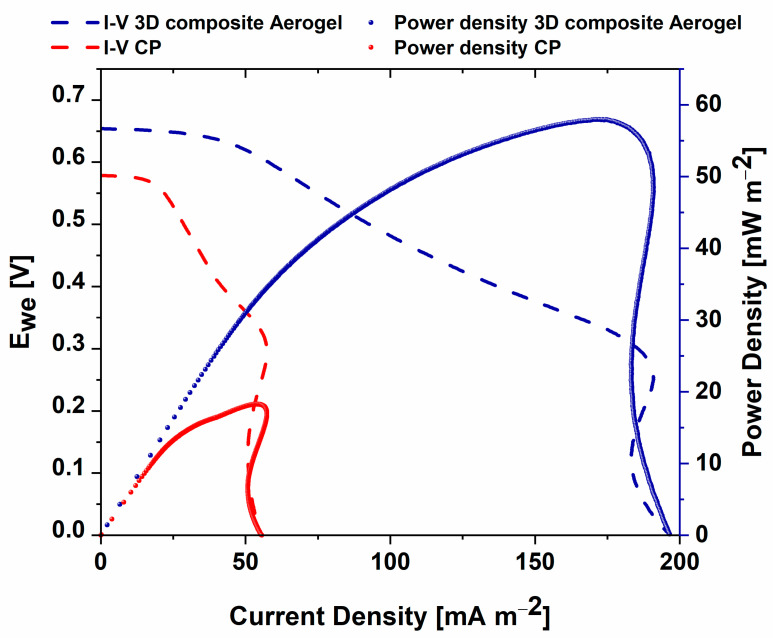
Linear sweep voltammetry curves obtained for *3D composite aerogels* and for the reference anode electrode. Blue dot and dash represent, respectively, power density and I–V trend for *3D-composite aerogel*, while red dot and dash report, respectively, power density and I–V trend for carbon-based material.

**Figure 6 nanomaterials-12-04335-f006:**
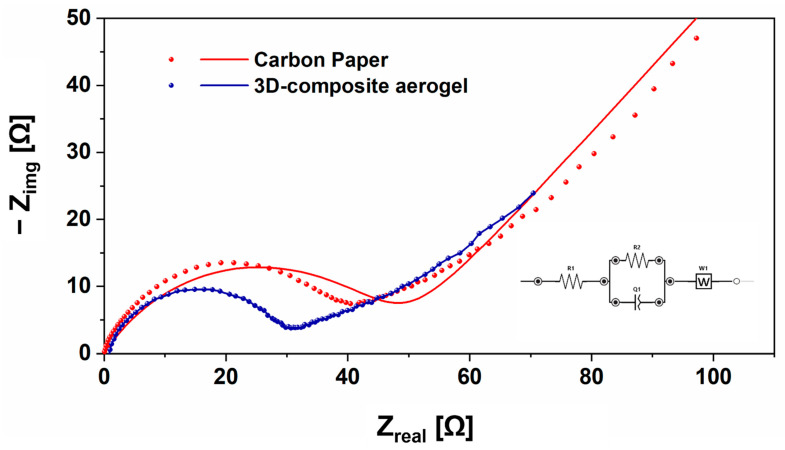
Electrochemical impedance spectroscopy of 3D composite aerogel (red dots represent the experimental data, red line reports the performed EIS fitting) compared with EIS of carbon paper (CP) (blue dots represent the experimental data and blue line the performed EIS fitting).

**Table 1 nanomaterials-12-04335-t001:** Several composite anode electrodes based on CNT investigated in the literature and developed as high-performing anode electrodes in microbial fuel cells (MFCs).

Composite Materials as Anode in MFCs	Overall Device Performance	Reference
Carbon nanotube (CNT)/polyaniline (PANI) was applied as composite anode	A maximum current density of 100 mA cm^−2^, in correspondence to which a maximum power density of 42 mW m^−2^, was achieved	[26]
A composite anode, based on carbon nanotube (CNT)/polyaniline (PANI), was involved to modify a stainless steel mesh	A maximum power density of 48 mW m^−2^ concerning anode performance	[27]
Composite anode made of polypyrrole-carboxymethyl cellulose (PPy-CMC) composite films to cover the nitrogen-doped carbon nanotubes sponge (N-CNTs)	A maximum volumetric power density of 4.88 W m^−3^	[28]
A composite electrode based on carbon black/stainless steel mesh (CB/SSM)	A maximum projected current density close to 10.07 ± 0.88 mA cm^−2^	[29]
3D porous carbon structure obtained by 3D printing technology with UV-curable resin. Final 3D structures were pyrolyzed at a temperature of 800 °C under N_2_ atmosphere	Achieved maximum power density of 233.5 ± 11.6 mWm^−2^	[33]
3D porous carbon sponges prepared through a pyrolysis treatment of nanostructured seitan composite, based on Fe-MIL-88B-NH2	A maximum power density, defined by normalizing the power output to internal device’s volume, was close to 11.21 W m^−3^ and a current density of 23.11 A m^−3^	[34]

## Data Availability

The raw/processed data required to reproduce these findings cannot be shared at this time as the data also form part of an ongoing study.

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
