# Peer review of "3D Composite PDMS/MWCNTs Aerogel as High-Performing Anodes in Microbial Fuel Cells"

_nanomaterials, 2022, doi:10.3390/nano12234335_

Round 1
Reviewer 1 Report
The manuscript of “3D composite PDMS/MWNCTs Aerogel as High-performing 1 Anodes in Microbial Fuel Cells” reports a 3D-composite aerogel based on PDMS, MWCNTs and commercial sugar, and as results, the SCMFCs with 3D-composite aerogel achieve current densities of (691.7±9.5) mA m-2, three or-21 ders of magnitude higher than the commercial carbon paper, (287.8±16.1) mA m-2. The creativity, organization, writing of this work is good, however, before accepting this manuscript, some revisions should be done to:
1. In the introduction, the authors reviewed recent developments of composite electrodes for SCMFCs, however, the value current density of efficient devices with high performances electrodes should be demonstrated and compared with the results of this work. It will further verify the creativity of this work.
2. The reasons that caused the improvements of performances are not clearly demonstrated, please give the clear relationship demonstrations of structure and properties. Also, authors can draw a mechanism schematic for the improvement of performances. “Nanomaterials 2020, 10, 944” can provide some mechanism schematic references and is meaningful to cite.
3. Can you provide the chemical component of commercial sugar and the source of it.
4. Did the authors optimize the fabrication conditions of the composite electrodes, for example, the component ratio? The structure of composite electrodes depends on the fabrication conditions, and the structure greatly affects the performances of devices. So it is meaningful to give clear relationship of fabrication condition, structure and performance.
Author Response
Dear Editor Teenie Zhu,
we would like to thank you and the Reviewers for the thorough evaluation and interest in the paper. Given below are the answers to the specific questions raised by the Reviewers and responses to their suggestions.
We provided to expand our literature survey as suggested and for the benefit of the Reviewers, all the changes, made in the revised version of the paper, are highlighted in red in the response and in the manuscript. Moreover, an English language editing was performed, and all comments were accepted.
We hope that the paper could now be suitable for publication.
Sincerely,
Giulia Massaglia
REVIEWER 1
The manuscript of “3D composite PDMS/MWNCTs Aerogel as High-performing 1 Anodes in Microbial Fuel Cells” reports a 3D-composite aerogel based on PDMS, MWCNTs and commercial sugar, and as results, the SCMFCs with 3D-composite aerogel achieve current densities of (691.7±9.5) mA m-2, three or-21 ders of magnitude higher than the commercial carbon paper, (287.8±16.1) mA m-2. The creativity, organization, writing of this work is good, however, before accepting this manuscript, some revisions should be done to:
- In the introduction, the authors reviewed recent developments of composite electrodes for SCMFCs, however, the value current density of efficient devices with high performances electrodes should be demonstrated and compared with the results of this work. It will further verify the creativity of this work
We appreciate this Reviewer’s comment that allowed comparing our results with the ones reached when composite electrodes for SCMFCs were investigated, leading thus to demonstrate how the results, reported in the present work, are comparable with other previous works. It is important to underline the main difference among most 3D composite electrodes proposed in the literature and our materials. To this purpose, indeed, it is vital to consider that 3D-composite aerogel, proposed in the present work, is a composite sample, showing thus a polymeric layer (Polydimethylsiloxane PDMS), biocompatible for microorganisms’ proliferation, which showed an insulator behaviour from an electric point of view and a conductive filler, carbon nanotubes (CNTs). In this scenario, CNTs play a key role not only to improve the electron transfer from microorganisms to electrode surface but also to induce an improved nanostructured habitat able to increase the bacterial growth, as mainly discussed in the literature [31]. Furthermore, differently from our 3D-composite aerogel, in the literature [27-35], all proposed and investigated composite anode electrodes were based on a conductive polymer, such as Polyaniline (PANI) [27-28], polypyrrole (PPy) [29], carbon black/stainless steel mesh composite electrode [30], or CNTs sponges [34-35]. Moreover, with the main purpose to demonstrate that overall performance, reached with 3D composite aerogel, resulted to be comparable with several optimized composite anode electrodes based on CNTs, a volumetric power density was defined by normalizing maximum achieved power output for the internal volume of our device (equal to 12.5 mL). SCMFCs with 3D-composite aerogel reached a maximum volumetric power density of (3.98±0.06) W m-3, resulting thus comparable with the performance achieved with various CNT-based anode materials. Indeed, as summarized in the Table 1, it is possible to appreciate how the overall performance, obtained with 3D-composite aerogel, results to be comparable with the one reached of various CNT-based anode materials.
Table 1 Several composite anode electrodes based on CNT investigated in the literature and developed as high performing anode electrodes in Microbial Fuel Cells.
|
Composite materials as anode in MFCs |
Overall Device Performance |
Reference |
|
Carbon nanotube (CNT)/polyaniline (PANI) was applied as composite anode |
A maximum current density of 100 mA cm-2 in correspondence of which a maximum power density of 42 mW m-2 was achieved |
[27] |
|
A composite anode, based on Carbon nanotube (CNT)/polyaniline (PANI), was involved to modify a stainless-steel mesh |
A maximum power density of 48 mW m-2 for what concerned anode performance |
[28] |
|
Composite anode made of polypyrrole-carboxymethyl cellulose (PPy-CMC) composite films to cover the nitrogen-doped carbon nanotubes sponge (N-CNTs) |
A maximum volumetric power density of 4.88 W m−3 |
[29] |
|
A composite electrode based on Carbon Black/stainless steel mesh (CB/SSM) |
A maximum projected current density close to 10.07 ± 0.88 mA cm-2
|
[30] |
|
3D porous carbon structure obtained by 3D printing technology with UV-curable resin. Final 3D structures were pyrolyzed at a temperature of 800°C under N2 atmosphere |
Achieved maximum power density of 233.5±11.6 mWm-2 |
[34] |
|
3D porous carbon sponges prepared through a pyrolysis treatment of nanostructured seitan composite, based on Fe-MIL-88B-NH2 |
A maximum power density, defined by normalizing the power output to internal device’s volume was close to 11.21 W m-3 and a current density of 23.11 A m-3 |
[35] |
We accordingly modified the main manuscript in the introduction section.
- The reasons that caused the improvements of performances are not clearly demonstrated, please give the clear relationship demonstrations of structure and properties. Also, authors can draw a mechanism schematic for the improvement of performances. “Nanomaterials 2020, 10, 944” can provide some mechanism schematic references and is meaningful to cite.
Among all possible technological components, such as electroactive bacteria species, device architecture, ion exchange membrane and organic substrates, bioelectrodes play a pivotal role in defining the MFCs’ performance [10-11]. Bioelectrodes, indeed, results to be the solid substrate on which the electroactive bacteria can proliferate, inducing, thus, the biofilm formation and the consequent exchange of electrons [12]. To guarantee these features. the physiochemical properties and structures of electrodes results to be a key factor that strictly affect the electron transfer at the biological/inorganic interface and define the maximum available surface area for electroactive bacteria’s attachment and growth. Carbon-based materials are considered one of the best- performing anode electrodes, able to combine the biocompatibility properties for microorganisms’ proliferation and the proper electrical conductivity, ensuring thus the electron transfer from electroactive bacteria and anode electrodes [9-10, 12-14]. The lower electrical conductivity of carbon-based materials, respect to the one of metal electrodes, was widely balanced by the best chemical surface and topographical features to favour biofilm growth and adhesion [15-16]. Moreover, carbon-based materials combine these properties with lightweight, high porosity, good chemical, mechanical and thermal stabilities, and low cost [17]. Previous studies applied different carbon materials as anode electrodes, such as carbon cloth, carbon paper graphite rod, carbon sponges and metal oxide foam with conductive coatings [18-19]. Among them, a particular interest for porous electrodes has arisen due to the high surface area capable to improve the bacteria growth, their adhesion and proliferation. Nevertheless, pore clogging frequently occurs during the phase of biofilm formation, when the porous electrodes were implemented. Consequently, the transport of substrates and waste products inside 3D-electrodes are largely limited, reflecting thus a low MFCs’ performance. With the main aim to overcome all these limits, in the present work, 3D-composite aerogel was proposed.
Our 3D composite aerogel, based on Polydimethylsiloxane (PDMS) and Multi Wall Carbon Nanotubes (MWCNTs), satisfied all mandatory features for anode electrodes, such as lightweight, good chemical and mechanical stability and high surface area and continuous porosity, suitable to offer an improved surface for bacterial proliferation and an enhanced mass transport. Indeed, in this scenario, PDMS was involved as biocompatible and flexible layer, able to improve bacteria proliferation on anode electrode, showing however, an insulating behaviour from an electrical point of view; while MWCNTs were employed as conductive filler suitable to create proper electron pathways to optimize the electron transfer from electroactive bacteria to anode surface, granting also the anode electrodes their unique properties and improving thus the electrochemical activity of microorganisms [33]. Moreover, as deeply investigated in the literature, during last years, CNTs played a key role not only to improve the electron transfer from microorganisms to electrode surface but also to induce an improved nanostructured habitat able to increase the bacterial growth. To this purpose, these nanostructured materials showed an emerging and interesting application as anodes in MFCs since they exhibit unique properties in terms of structural features, mechanical, electrical, and physical and chemical characteristics [32-33]. CNTs resulted to be fundamental to improve overall MFCs’ performance thanks to their capability to enhance the electrochemical activity of microorganisms [33-35] and to create a proper electrons pathway, enhancing thus their transfer from microorganisms to anode surface.
In line with all these considerations, an anode electrode, which satisfies all presented properties, allowed the biofilm formation, where electroactive bacteria can directly transfer the produced electrons to anode surface, leading thus to reach an anode material suitable to improve the electrochemical activity of microorganisms. The Figure, reported in the attached file, highlights the direct electron transfer, which can be exploited through a direct contact of microorganisms’ cytochromes and the anode surfaces or towards nanowires, directly produced by bacteria, connecting themselves and the anode surface. This scheme is proposed in line with the Reviewer’s suggested paper [Nanomaterials 2020, 10, 944].
This change was applied in the Figure 1 of main manuscript and reference citation was reported accordingly.
- Can you provide the chemical component of commercial sugar and the source of it.
We thank the Reviewer for this request that help us to better specify the chemical composition of sugar. As commonly known, chemically sugar is based on carbon (C), oxygen (O) and hydrogen (H) atoms, bonded one with each other and it is classified as a carbohydrate. In the present work, in particular, a commercial sugar, was used as removable template, since it presents the correct granularities’ dimension suitable to create, modulate high continuous porosity inside the final 3D samples. To this purpose, we involved a refined form of sucrose, representing the primary example of a disaccharide. The manuscript was modified accordingly
- Did the authors optimize the fabrication conditions of the composite electrodes, for example, the component ratio? The structure of composite electrodes depends on the fabrication conditions, and the structure greatly affects the performances of devices. So it is meaningful to give clear relationship of fabrication condition, structure and performance.
We especially thanks to the Reviewer for this consideration, highlighting a mistake done during the definition of MWCNTs’ amount during the synthesis of 3D-composite aerogel. To better explain the synthesis of 3D composite aerogel, it was important to underline the effective role of all components involved to obtain final samples. Firstly, we prepared a mixture based on Polydimethylsiloxane (PDMS) and its curing agent with a ratio of 10:1, which represents the standard ratio for both components, leading thus to provide desirable mechanical properties, in terms of flexibility and an optimum biocompatibility for electroactive bacteria growth. Successively, we added the minimum amount of MWCNTs, close to 10wt% defined in relation with the amount of PDMS, needed to improve the electrical conductivity of final samples, in line with all work reported in the literature [27-35]. Finally, all mixtures were poured over the sugar, allowed them infiltrating into the porous structures thanks to capillary force. Consequently, the size of the pores and porosity was directly affected and tuned by the granularity of sugar. The final 3D-composite aerogel was obtained towards a curing process, implemented in ambient condition at a temperature of 100°C for 15 min and the sugar was subsequently dissolved in a hot water under sonication bath for 1h. One of advantages to implement this synthesis process can be identified by the fact that the final features of 3D composite aerogel, in terms of electrical conductivity and tunable continuous porosity, are not dependent onto all those parameters, which are really difficult to control (i.e. environmental condition). On the contrary, obtained properties of 3D-composite aerogel are strictly correlated with the granularity dimension of sugar and amount of MWCNTs, representing controllable factors.

Reviewer 2 Report
There are some weaknesses through the manuscript which need improvement. Therefore, the submitted manuscript cannot be accepted for publication in this form, but it has a chance of acceptance after a major revision. My comments and suggestions are as follows:
1- Abstract gives information on the main feature of the performed study, but some details about the conducted tests must be added. However, a concise abstract is needed.
2- Authors must clarify necessity of the performed research. Aims and objectives of the study, and also differences with the previous review papers must be clearly mentioned.
3- The literature study must be enriched. For instance, authors can read and refer to the recently published relevant papers: (a) https://doi.org/10.1007/s40094-016-0217-9 (b) https://doi.org/10.1016/j.microrel.2018.10.008
4- It would be beneficial to the reader if a figure showing the overall structure of the paper to enhance the readability of the paper.
5- Appropriate reference is needed for some sentences. For example, last sentence of the first paragraph in introduction.
6- The main reference of each formula must be cited. All figures must be depicted in a high quality. Font size of the formula must be changed.
7- Since it is an experimental study, authors must add real figures to show tests (e.g., section 2.2. and 2.3).
8- Standard deviation in the obtained results must be discussed.
9- In its language layer, the manuscript should be considered for English language editing. There are sentences which have to be rewritten.
10- The conclusion must be more than just a summary of the manuscript. List of references must be updated based on the proposed papers. Please provide all changes by red color in the revised version.
Author Response
Dear Editor Teenie Zhu,
we would like to thank you and the Reviewers for the thorough evaluation and interest in the paper. Given below are the answers to the specific questions raised by the Reviewers and responses to their suggestions.
We provided to expand our literature survey as suggested and for the benefit of the Reviewers, all the changes, made in the revised version of the paper, are highlighted in red in the response and in the manuscript. Moreover, an English language editing was performed, and all comments were accepted.
We hope that the paper could now be suitable for publication.
Sincerely,
Giulia Massaglia
REVIEWER 2
There are some weaknesses through the manuscript which need improvement. Therefore, the submitted manuscript cannot be accepted for publication in this form, but it has a chance of acceptance after a major revision. My comments and suggestions are as follows:
1- Abstract gives information on the main feature of the performed study, but some details about the conducted tests must be added. However, a concise abstract is needed.
We modified the abstract with the aim to give more information on the main feature of performed study in a more concise way.
2- Authors must clarify necessity of the performed research. Aims and objectives of the study, and also differences with the previous review papers must be clearly mentioned.
We appreciate this Reviewer’s comment that allowed comparing our results with the ones reached when composite electrodes for SCMFCs were investigated, leading thus to demonstrate how the results, reported in the present work, are comparable with other previous works [27-35]. We added a Table that allowed summarizing overall performance, obtained with our 3D composite aerogel and one reached with various CNTs-based anode materials. As summarized in the Table, it was possible to appreciate how SCMFCs’ performance with 3D composite aerogel resulted to be comparable with the one of various composite anode electrode investigated in the literature [27-35].
|
Composite materials as anode in MFCs |
Overall Device Performance |
Reference |
|
Carbon nanotube (CNT)/polyaniline (PANI) was applied as composite anode |
A maximum current density of 100 mA cm-2 in correspondence of which a maximum power density of 42 mW m-2 was achieved |
[27] |
|
A composite anode, based on Carbon nanotube (CNT)/polyaniline (PANI), was involved to modify a stainless-steel mesh |
A maximum power density of 48 mW m-2 for what concerned anode performance |
[28] |
|
Composite anode made of polypyrrole-carboxymethyl cellulose (PPy-CMC) composite films to cover the nitrogen-doped carbon nanotubes sponge (N-CNTs) |
A maximum volumetric power density of 4.88 W m−3 |
[29] |
|
A composite electrode based on Carbon Black/stainless steel mesh (CB/SSM) |
A maximum projected current density close to 10.07 ± 0.88 mA cm -2
|
[30] |
|
3D porous carbon structure obtained by 3D printing technology with UV-curable resin. Final 3D structures were pyrolyzed at a temperature of 800°C under N2 atmosphere |
Achieved maximum power density of 233.5±11.6 mWm-2 |
[34] |
|
3D porous carbon sponges prepared through a pyrolysis treatment of nanostructured seitan composite, based on Fe-MIL-88B-NH2 |
A maximum power density, defined by normalizing the power output to internal device’s volume was close to 11.21 W m-3 and a current density of 23.11 A m-3 |
[35] |
It is important to underline the main difference among most 3D composite electrodes proposed in the literature and our materials. To this purpose, indeed, it is vital to consider that 3D-composite aerogel, proposed in the present work, is not a carbon-based material, but a composite sample, showing thus a polymeric layer, biocompatible for microorganisms’ proliferation, which showed an insulator behaviour from an electric point of view and a conductive filler, carbon nanotubes (CNTs). In this scenario, CNTs play a key role not only to improve the electron transfer from microorganisms to electrode surface but also to induce an improved nanostructured habitat able to increase the bacterial growth [27-35]. Furthermore, differently from our 3D-composite aerogel, in the literature [31], all proposed and investigated composite anode electrodes were based on a conductive polymer, such as Polyaniline (PANI) [27-28], polypyrrole (PPy) [29], carbon black/stainless steel mesh composite electrode [30], or CNTs sponges [34-35].
With the main purpose to clarify our performed research, we modified the manuscript accordingly.
3- The literature study must be enriched. For instance, authors can read and refer to the recently published relevant papers: (a) https://doi.org/10.1007/s40094-016-0217-9 (b) https://doi.org/10.1016/j.microrel.2018.10.008
We thank the reviewer to highlight the presence of these studies in the literature. For what concerned the research work [10.1016/j.microrel.2018.10.008], Yim et al. focused their attention onto the influence of MWCNTs concentration on the thermodynamic and mechanical reliability properties of anisotropic conductive adhesives (SACAs), leading thus to confirm that CNT-filled SACA achieved much superior thermos-mechanical reliability properties, such as electrical and mechanical reliability performances, respect to the ones obtained by SACA without MWCNTs. Although this composite material is totally different from the one proposed in our works, this research paper allows us to strengthen the concept that MWCNTs improve different properties of final nanostructured material, giving it all required features to achieve a high-performing anode electrode. However, for what concerned the second suggested paper [https://doi.org/10.1007/s40094-016-0217-9], it is not properly fitting with the main purpose of our research paper.
4- It would be beneficial to the reader if a figure showing the overall structure of the paper to enhance the readability of the paper.
We thank the Reviewer for this consideration, and with the main aim to enhance the readability of the paper, we modified the Figure 1 in the main manuscript, as represented here:
5- Appropriate reference is needed for some sentences. For example, last sentence of the first paragraph in introduction.
Thanks to the Reviewer. We added properly all needed reference in the main manuscript
6- The main reference of each formula must be cited. All figures must be depicted in a high quality. Font size of the formula must be changed.
The definition of the porosity, considering the density of 3D-composite aerogel ρ_(3D-areogel) and the density of a bulk substrate made of the same materials, ρ_bulk, according to the Equation 1, was reported in our previous work [39].
[(1-(ρ_(3D-areogel) ⁄ ρ_bulk )*100]
In this work, the authors demonstrated the tunability of porosity of nanofibers mats strictly correlated with the elettrospinning process. Similar to these involved nanofibers, 3D-composite aerogels can be considered as nanostructure material, whose porosity can be affected by granularity of sugar and the presence of MWCNTs inside the samples.
We modified the main manuscript as suggested by Reviewer. We hope that following those suggestions, all figures will be depicted in a high quality and the font size of the formula will result to be well presented.
7- Since it is an experimental study, authors must add real figures to show tests (e.g., section 2.2. and 2.3).
Following previous Reviewer’s suggestion, expressed in the point 4, we modified the Figure 1 in the main manuscript with the main purpose to improve the readability of the paper and at the same time to show all performed tests toward real figures
8- Standard deviation in the obtained results must be discussed.
Standard deviation in the obtained results was calculated by considering all available data, that we collected during whole experimental period. For what concerning the definition of overall SCMFCs’ performance, indeed, for each anode electrode, we implemented 2 SCMFCs and experimental data were continuously recorded for a whole period longer than 1 month. Figure 4 allowed appreciating the repeatability of maximum current density reached with both of anode electrodes, 3D-composite aerogel, and CP, respectively, leading thus the opportunity to evaluate the dispersion of these experimental data from its mean. It is important to specify that Figure 4 reports only the mean values of current density versus the experimental period. The same consideration can be applied onto electrical conductivity and porosity definitions, since we repeated 3 times the characterizations, implemented for 10 samples for each proposed structure. We modified the main manuscript with the purpose to better specify the functional properties of our samples, as following:
“Figure 3 allowed confirming how the aerogel structure presented a higher electrical conductivity, close to (26.5 ± 2.8) mS cm-1, which results to be two times higher than the one of (10.6 ± 2.5) mS cm-1 obtained with 3D composite bulk samples. However, on the contrary, 3D-composite electrical conductivity results to be comparable with the one reached when carbon paper (CP, (23.6 ± 2.5) mS cm-1) was used as reference anode electrode, commonly characterized by high performance in terms of electrical conductivity”
9- In its language layer, the manuscript should be considered for English language editing. There are sentences which have to be rewritten.
We appreciated the comment of the Reviewer, and we modified the document following the detailed instructions and comments of Reviewer. We hope that following those suggestions, the new version will result to be better organized and the experiments well presented.
10- The conclusion must be more than just a summary of the manuscript. List of references must be updated based on the proposed papers. Please provide all changes by red color in the revised version.
The conclusion was accordingly modified and, moreover, we provided to expand our literature survey as suggested

Round 2
Reviewer 1 Report
Authors has improved the quality of the manuscript greatly. And Figure 1 were added the scheme based on the reference of [Nanomaterials 2020, 10, 944], please cite this reference.
Author Response
Authors has improved the quality of the manuscript greatly. And Figure 1 were added the scheme based on the reference of [Nanomaterials 2020, 10, 944], please cite this reference.
We thanks the Reviewer for this comment. The reference was accordingly added to main manuscript as followed:
[44] Paczesny J., Bielec K. Reivew: Application of Bacteriophages in Nanotechnology. Nanomaterials 2020, 10, 944
Reviewer 2 Report
The revised version of the manuscript appears to be suitable for publication.
Author Response
Thank you very much for appreciating the effort, spent during revision process